# Assessing the Orthogonality of Phage-Encoded RNA Polymerases for Tailored Synthetic Biology Applications in *Pseudomonas* Species

**DOI:** 10.3390/ijms24087175

**Published:** 2023-04-13

**Authors:** Eveline-Marie Lammens, Nathalie Feyaerts, Alison Kerremans, Maarten Boon, Rob Lavigne

**Affiliations:** Laboratory of Gene Technology, Department of Biosystems, KU Leuven, Kasteelpark Arenberg 21 Box 2462, 3001 Leuven, Belgium

**Keywords:** *Pseudomonas*, T7-like phages, RNA polymerase, lysozyme, orthogonality

## Abstract

The phage T7 RNA polymerase (RNAP) and lysozyme form the basis of the widely used pET expression system for recombinant expression in the biotechnology field and as a tool in microbial synthetic biology. Attempts to transfer this genetic circuitry from *Escherichia coli* to non-model bacterial organisms with high potential have been restricted by the cytotoxicity of the T7 RNAP in the receiving hosts. We here explore the diversity of T7-like RNAPs mined directly from *Pseudomonas* phages for implementation in *Pseudomonas* species, thus relying on the co-evolution and natural adaptation of the system towards its host. By screening and characterizing different viral transcription machinery using a vector-based system in *P. putida*., we identified a set of four non-toxic phage RNAPs from phages phi15, PPPL-1, Pf-10, and 67PfluR64PP, showing a broad activity range and orthogonality to each other and the T7 RNAP. In addition, we confirmed the transcription start sites of their predicted promoters and improved the stringency of the phage RNAP expression systems by introducing and optimizing phage lysozymes for RNAP inhibition. This set of viral RNAPs expands the adaption of T7-inspired circuitry towards *Pseudomonas* species and highlights the potential of mining tailored genetic parts and tools from phages for their non-model host.

## 1. Introduction

Due to their long-standing co-evolution, bacteriophage genomes encode unique biological parts and cell modulators that are integrally adapted to their host, including viral RNA polymerases (RNAPs) and orthogonal promoters [1]. The main example hereof is the transcriptional machinery of coliphage T7. Since its discovery over 30 years ago, the T7 transcription elements have become indispensable for both high-yield protein production in *Escherichia coli* and the construction of complex Synthetic Biology (SynBio) circuitry [2,3]. The transcriptional machinery of T7, including its small, single subunit, RNAP, 17 bp promoter (P_T7_), and specific T7 lysozyme, has since been commercialized by Novagen (Merck Group, Darmstadt, Germany) as the well-known pET system. The pET system consists of three modules and can be induced with isopropyl-β-D-thiogalactopyranoside (IPTG), which drives the expression of the T7 RNAP by the LacI expression system (Figure 1). Next, the T7 RNAP transcribes any gene of interest from P_T7_ to extremely high levels, resulting in a high (recombinant) protein yield. Due to the strong activity of the T7 RNAP, strict regulation of this system is required to avoid the unwanted expression of the gene of interest in pre-production cultures. The presence of the T7 lysozyme supports this regulation, as it inhibits the T7 RNAP when expressed in low basal amounts prior to IPTG induction.

Apart from the traditional pET system for protein production, the T7 RNAP has several interesting properties which have been exploited in SynBio applications. First, the exceptional transcriptional activity of the T7 RNAP can be implemented in applications beyond protein production, such as signal amplification in biosensors and enzymatic activity assays [4,5]. Second, the modular structure of the T7 RNAP allows the splitting of the T7 RNAP gene into two parts to create AND and OR gates in synthetic circuits [6,7,8,9,10]. In addition, the separate T7 RNAP modules can be fused to other enzymes, such as deaminases, to enable base-editing of a target sequence [11]. Third, the T7 RNAP solely recognizes P_T7_ and can, therefore, function fully orthogonally from the host’s transcriptional machinery. 

In the past decade, several bacterial species beyond *E. coli* have emerged as valuable hosts for biotechnological and SynBio applications. In specific cases, non-model bacteria are better tailored to produce certain proteins or metabolites due to their unique metabolism, ability to thrive in harsh bioreactor setups, and/or intrinsic resistance to toxic proteins and metabolites that are industrially relevant [1]. These species include members of the *Pseudomonas* genus, with a special focus on the metabolically versatile and robust *P. putida* [12,13]. As such, SynBio parts and tools, including a high-yield expression system, for these hosts are absolutely essential. The powerful pET system would be a great addition to the *Pseudomonas* toolbox but performs sub-par in *Pseudomonas* hosts as it is based on coliphage T7 and is further optimized towards *E. coli*. In *P. putida*, induction of the T7 RNAP and T7 lysozyme results in arrested cell growth, leading to high cell burden and increased mutational pressure [7,14,15,16,17]. Moreover, the LacI system, which is often used to express the T7 RNAP, is regulated differently in *P. putida* and causes extremely high levels of T7 RNAPs under uninduced conditions [18,19,20]. In the past, attempts have been made to solve these issues by employing the XylS/*Pm* system instead of LacI [5,17] or by using antisense RNA to tightly regulate the system [15]. However, these systems still rely on the toxic T7 RNAP. Furthermore, Liang et al. reported a functional alternative RNAP for *P. putida* but did not report on the fitness of the expression strains [21]. As a result, the pET system is rarely used in *P. putida*, and researchers instead rely on the established XylS/*Pm* and RhaRS/*P_rhaBAD_* expression systems for protein production. These expression systems are generally well-regulated and characterized in *P. putida* but lack the extremely high transcription levels enabled by the T7 RNAP. 

In this work, we characterize four alternative RNAPs from *Pseudomonas* phages that exhibit a broad range of expression levels, do not show any toxicity to their host, and operate orthogonally to each other and the T7 RNAP. Efficient expression of these RNAPs was achieved using the XylS/*Pm* system, and tight regulation was improved by introducing the corresponding phage lysozymes. 

## 2. Results

### 2.1. T7-like Pseudomonas Phage Genomes Encode Putative RNA Polymerases, Lysozymes, and Phage-Specific Promoters

Previous work has shown that the T7 RNAP causes significant growth deficits in *Pseudomonas* cultures upon expression [7,14,15,16,17]. To reduce this cytotoxicity for *Pseudomonas*, one could employ two strategies: (1) optimize the T7 RNAP with directed evolution for *Pseudomonas* or (2) identify novel and optimized phage RNAPs from *Pseudomonas* phages. In this work, we will opt for the latter and focus on exploring the existing diversity of RNAPs among *Pseudomonas* phages for reduced cytotoxicity and increased transcriptional activity. While all *Autographiviridae* phages typically encode a viral RNAP, our analysis focused on T7-like phages, as they generally encode a small, single-subunit RNAP with clearly delineated promoter recognition sequences [7,22]. To date, 36 T7-like *Pseudomonas* phages have been isolated and fully sequenced (Appendix A). Based on the sequence alignment of these phage RNAPs, they can be subdivided into eleven distinct clades. We chose four phages from different clades (PPPL-1, Pf-10, 67PfluR64PP, and phi15) and analyzed their annotated RNAP, lysozyme, and predicted-consensus promoter with several toxicity- and fluorescence-based assays. The genomic organization of phages T7, phi15, PPPL-1, Pf-10, and 67PfluR64PP and their promoter region preceding the major capsid protein (MCP) is illustrated in Figure 2.

### 2.2. Screening Non-Toxic Phage RNA Polymerases and Their Transcriptional Activity

First, the four phage RNAPs were screened for low cytotoxicity in *P. putida* and *P. aeruginosa*, compared to the T7 reference model. As the *Pseudomonas* phages co-evolved with their host, we hypothesized that their early-expressed RNAP should have primed efficient production of viral particles and not trigger the host’s toxicity. To confirm this hypothesis, the RNAP from our selected *Pseudomonas* phages, Pf-10, phi15, PPPL-1, and 67PfluR64PP, were cloned into pSTDesX, introduced in either *P. putida* or *P. aeruginosa* and induced with 1 mM 3mBz (3-methylbenzoate) from the XylS/*Pm* expression system. This expression system is considered the golden standard for *P. putida* and has successfully driven T7 RNAP expression in previous research to circumvent LacI-related regulatory issues in *Pseudomonas* [5,17]. As anticipated, the T7-like RNAPs did not significantly reduce the final OD_600_ of the host after 12 h of induction, with the exception of some limited growth reduction induced by the expression of the 67PfluR64PP RNAP in *P. aeruginosa* (Tukey HSD, *p* < 0.001) (Figure 3a). By contrast, the T7 RNAP caused a significant growth stop and growth retardation in both species, as anticipated (Tukey HSD, *p* < 0.0001) (Figure 3a and Appendix A). 

It should be noted that to include the T7 RNAP as a positive control in further assays, its inducer concentration should be reduced from 1 mM to 0.3 mM 3mBz in subsequent experiments to limit the toxic effect. Next, the transcriptional activity of the RNAPs was assessed indirectly by measuring the level of the msfGFP (monomeric superfolder green fluorescent protein) fluorescence from a *phage promoter-msfGFP* reporter construct. The predicted phage promoters and 5′ untranslated regions (UTRs) from the phages’ major capsid protein (MCP) are shown in Figure 2b and were cloned upstream of the *msfGFP* gene in the pBGDes. The reporter construct in the pBGDes was genomically integrated as a single copy in the host’s Tn7 *attB* site to limit noise from the copy number differences. These reporter constructs were first tested individually in the host in the absence of the phage RNAPs to confirm whether the phage promoters are not recognized by the host RNAP, which would be indicated by a lack of msfGFP expression (Figure 3b). In both *P. putida* and *P. aeruginosa*, no significant gene expression was observed from any of the phage promoters by the host RNAP, except for the T7 promoter. This result suggests that the T7 promoter can be recognized by the host RNAP (albeit very weakly) and is therefore not fully orthogonal to the host RNAP in *Pseudomonas*, contrary to previous assumptions [17].

Next, all of the phage RNAPs were introduced in the corresponding reporter strains, and msfGFP expression was monitored for 12 h in the absence and presence of 0.3 mM of a 3mBz inducer. All of the tested phage RNAPs displayed significant transcriptional activity in *P. putida* after 12 h of induction (pairwise Wilcoxon, *p* < 0.001), whereas only T7, phi15, and PPPL-1 RNAP produced significant msfGFPs in *P. aeruginosa* (pairwise Student’s *t*-test, *p* < 0.05 (T7, phi15) and *p* > 0.05 (PPPL-1); individual Student’s *t*-tests for PPPL-1 vs. NC, *p* < 0.05) (Figure 3c). The reason for the apparent inactivity of Pf-10 and 67PfluR64PP RNAP in *P. aeruginosa* is unclear but could be due to improper expression of these RNAPs in this species, as the *RNAP* genes were not codon-optimized to the respective hosts. In the current setup, the T7 RNAP generated the highest msfGFP production per cell of 1185 nM and 1972 nM 5(6)-FAM/OD_600_ in *P. putida* and *P. aeruginosa* after 12 h of induction, followed by 543 nM and 512 nM 5(6)-FAM/OD_600_ for phi15 RNAP, respectively. In *P. putida*, the phage RNAPs of PPPL-1, Pf-10, and 67PfluR64PP showed reduced expression levels. Taken together, these less-active enzymes provide an RNAP library together with T7 and phi15 RNAP, covering a broad range of transcriptional expression. In this regard, it should be noted that while the transcriptional activity of T7 RNAP remained highest in this experiment, its cytotoxicity and the difficulty this brings for cloning, handling, and expressing the T7 RNAP in *Pseudomonas* should be considered. As such, a non-toxic RNAP with more average expression levels like the phi15 RNAP may still prove preferable for many applications.

Looking back at the MCP promoter region of the phages (Figure 2b), extensive conservation of the promoter motif can be observed, which could indicate the cross-recognition of the promoters by the phage RNAPs. To test if any cross-recognition occurred, all 25 combinations of the T7, phi15, PPPL-1, Pf-10, and 67PfluR64PP RNAPs and promoters were set up in *P. putida* and induced with 0.3 mM 3mBz. Surprisingly, nearly full orthogonality was observed between the phage RNAPs, as illustrated in Figure 4 (Appendix A). Except for the phage RNAP 67PfluR64PP, all of the RNAPs yielded high msfGFP levels from their native promoter, whereas the msfGFP levels originating from other phage promoters remained insignificant (pairwise Student’s *t*-test, *p* > 0.05). Only for the phi15 RNAP was minimal cross-recognition observed from the T7 promoter (86 5(6)-FAM/OD_600_), while this effect was not observed for the T7 RNAP in combination with the phi15 promoter. As such, this set of phage RNAPs and promoters can not only be used to build synthetic AND gates and resource allocators [6,8], but the combination of the T7 and phi15 RNAPs with the T7 promoter can even allow the construction of an OR gate and many other setups.

### 2.3. Phages phi15, PPPL-1, Pf-10, and 67PfluR64PP Encode Short, 17 bp Promoters

The T7 promoter is a well-characterized 17 bp sequence with an N-terminal AT-rich recognition loop (−17–13), a specificity loop of 5 bp (−11–7), and an unwinding region (−4–1) (Figure 5 and Appendix A) [22,23,24,25,26]. The predicted phage promoters of PPPL-1, Pf-10, 67PfluR64PP, and phi15 all show a high sequence similarity to the T7 promoter. As such, it is reasonable to assume that these promoters contain a similar structure. To validate the exact transcription start site (TSS) of the phage promoters in vivo, a 5′-capping-RACE experiment was performed using the *P. putida* KT2440 strains pA0RA0, pB0RB0, pC0RC0, pD0RD0, and pE0RE0. 5′-capping-RACE (Rapid Amplification of cDNA Ends) allows the capture of full-length mRNA molecules. Subsequent sequencing of their 5′ termini precisely determines the TSSs [27].

The capping-RACE experiment confirmed the start sites of the predicted promoters of phi15, PPPL-1, Pf-10, and 67PfluR64PP. Overall, these T7-like promoters showed a canonical length of 17 bp (18 bp for P_Pf-10_) and two AT-rich regions flanking the presumed recognition loop (Figure 5 and Appendix A). This validation allowed us to successfully pair the promoters with other 5′ UTRs, including BCD2, a standardized, highly-potent UTR with a bicistronic design commonly used in *P. putida* (Appendix A) [28,29]. 

These standardized bicistronic UTRs are of specific interest for the use of these phage promoters in synthetic circuitry. Indeed, BDC2 and other bicistronic designs have the advantage of circumventing the well-known problem of secondary structure formation between the RBS and the downstream gene of interest, potentially inhibiting proper translation [28,30]. This allows the user to reliably reuse the expression construct in a standardized design for different genes of interest without the need for individually optimized 5′ UTRs for each construct.

### 2.4. T7-like Phage Lysozymes Inhibit Transcriptional Activity of Their Corresponding Phage RNAP

Due to the strong transcriptional activity of T7-like phage RNAPs, a limited production of the phage RNAP can rapidly lead to significant expression levels of the reporter gene in uninduced conditions. This observation can also be made for the uninduced *P. putida* samples from the previous assay, where all strains except the 67PfluR64PP RNAP produced msfGFP in significantly higher concentrations compared to the negative control (Tukey HSD, *p* < 0.05) (Figure 3c). The highest levels of leaky expression in *P. putida* are observed for phi15 (160 5(6)-FAM/OD_600_) and T7 (120 5(6)-FAM/OD_600_). This leaky msfGFP expression is likely caused by the low basal expression of the phage RNAP from the XylS/*Pm* expression system [18]. To limit this leaky expression, the corresponding phage lysozyme can be added to the system to inhibit the phage RNAP and therefore decrease the transcription of the reporter gene, analogous to previous work [3,31]. As indicated in Figure 2, the genomes of phi15, PPPL-1, Pf-10, and 67PfluR64PP all encode an early-expressed lysozyme with high similarity to the T7 lysozyme. This lysozyme could potentially inhibit the corresponding phage RNAP but could also exhibit cytotoxicity due to their intrinsic amidase activity, as shown by the overexpression of the T7 lysozyme [3]. Therefore, all of the lysozymes were cloned into pSTDesR with the RhaRS/*P_rhaBAD_* expression system, introduced into *P. putida* KT2440 and *P. aeruginosa* PAO1, induced with 10 mM Rha (rhamnose), and screened for the host’s toxicity. Interestingly, all of the phage lysozymes significantly reduced the cell growth of *P. aeruginosa* (Tukey HSD, *p* < 0.01), while in *P. putida*, no significant toxicity was observed from the expression of the phi15, PPPL-1, and Pf-10 lysozymes after induction (Tukey HSD, *p* > 0.1) (Figure 6a). This is in contrast to the moderate toxicity observed by the T7 and 67PfluR64PP lysozymes, respectively (Tukey HSD, *p* < 0.01) (Figure 6a). The rhamnose induction concentration in further assays will be reduced to 5 mM instead of 10 mM to limit the toxic effect but still allow sufficient lysozyme expression to inhibit the RNAP, as determined for the T7 system (Appendix A).

Next, the inhibitory effect of the lysozymes on the phage RNAP was analyzed in *P. putida* and *P. aeruginosa* by introducing the phage lysozyme, RNAP, and *phage promoter-msfgGFP* reporter construct in the host and monitoring the msfGFP output after induction with 4 mM Rha. Upon the induction of lysozyme expression, all *P. putida* samples showed a significantly reduced msfGFP output compared to their uninduced counterparts (Figure 6b), indicating that all of the lysozymes successfully inhibited their corresponding RNAP and reduced leaky expression. The largest reductions in msfGFP were observed for the Pf-10 (−80%) and T7 (−79%) systems, followed by a medium reduction for the 67PfluR64PP (−51%) and PPPL-1 (−39%) systems. The phi15 lysozyme caused only a −30% reduction of the msfGFP output, therefore still displaying a significant level of leaky fluorescence of 77 nM 5(6)-FAM/OD_600_. In *P. aeruginosa,* on the other hand, a slight reduction trend in the leaky expression was observed for T7, phi15, and PPPL-1 upon the lysozyme expression, but these reductions did not prove to be significant (one-sided Student’s *t*-test, *p* > 0.05). Therefore, the assay was repeated with 5 mM Rha instead of 4 mM to increase the lysozyme expression without causing significant growth retardation. As displayed in Figure 6b, the msfGFP output of the induced *P. aeruginosa* strains now trended lower than the uninduced controls in all of the strains, but this was only statistically significant for the phi15 sample (one-sided Student’s *t*-test, *p* < 0.05). Overall, these results indicate that the predicted phage lysozymes can reduce basal msfGFP expression levels originating from the phage RNAPs and should be introduced to improve the stringency of the expression circuitry.

### 2.5. T7-like Phage Lysozymes Efficiently Inhibit Phage RNAPs from Related T7-like Phages

The viral RNAP from phage phi15 resulted in high expression levels in *P. putida* and *P. aeruginosa* (Figure 3) but also exhibited significant leakiness, even in the presence of the phi15 lysozyme (Figure 6b). In an attempt to further reduce the leaky expression, the rhamnose concentration was increased up to 100 mM in *P. putida*, to no avail (Appendix A). As shown for the T7 system, the lysozyme inhibitory action stems from its N-terminal tail, with which it binds to the phage RNAP and causes allosteric inhibition of this enzyme [23,32,33]. Changes in the N-terminal tail sequence can, therefore, result in either weaker or stronger binding of the RNAP, which can, in turn, influence the inhibitory activity. The five studied phage lysozymes in this work show a range of inhibitory performances and encode distinct N-terminal regions (Appendix A). Therefore, we paired the lysozymes of T7, phi15, PPPL-1, Pf-10, and 67PfluR64PP with the phi15 RNAP and *Pphi15-msfGFP* reporter construct to test whether these lysozymes also have different abilities to bind and inhibit the phi15 RNAP. The resulting *P. putida* strains were induced with 4 mM rhamnose to express the lysozyme, after which the msfGFP output was measured for 12 h (Figure 7a). Despite a significant growth reduction caused by the 67PfluR64PP and Pf-10 lysozymes upon induction, similar to the previous assay, these lysozymes showed a remarked reduction in the leakiness caused by the phi15 RNAP. The 67PfluR64PP lysozyme reduced the leaky expression by −58%. Interestingly, the Pf-10 lysozyme even significantly outperformed the phi15 lysozyme with an −84% decrease in the msfGFP output (Tukey HSD, *p* < 0.0001).

To verify that this striking result was due to the unique N-terminal region of the Pf-10 lysozyme and not to any other amino acid differences between the sequences of the Pf-10 and phi15 lysozymes (or even due to the slight toxicity of the Pf-10 lysozyme), a phi15 lysozyme mutant was engineered in which the first nine N-terminal amino acids were substituted for the first ten amino acids of the Pf-10 lysozyme while maintaining the other 145 amino acids of the phi15 lysozyme. The resulting *P. putida* strain with the phi15 RNAP, reporter construct, and phi15 lysozyme (AA1-9 > Pf-10(AA1-10)) showed no reduction in cell growth and generated a fluorescent output that was nearly identical to the sample with the Pf-10 lysozyme when induced with 5 mM rhamnose (Tukey HSD, *p* > 0.05) (Figure 7b). This confirms the determining role of the N-terminal region on RNAP inhibition and strongly suggests that the N-terminal region of the Pf-10 lysozyme allows a stronger binding interaction with the phi15 RNAP than the phi15 lysozyme. To pinpoint the exact amino acids causing the increased inhibitory activity, five additional phi15 lysozyme mutants were created and introduced in *P. putida* with the phi15 RNAP and reporter construct: phi15 lysozyme (G3R), (G3Q), (G3RQ), (K5Q), and (K7N,E8K). Except for the G3Q mutant (*p* < 0.01), none of the mutants had a significant impact on cell growth.

While the phi15 lysozyme mutants (G3Q), (K5Q), and (K7N,E8K) did not improve the inhibitory activity of the lysozyme (Figure 7b), mutants (G3R) and (G3RQ) did decrease the leaky expression to 57 nM and 11 nM 5(6)-FAM/OD_600_, respectively. This is a significant improvement compared to the 122 nM 5(6)-FAM/OD_600_ observed for the wild-type phi15 lysozyme. Moreover, the results of the G3RQ mutant even outperformed those of the phi15 lysozyme (AA1-9 > Pf-10(AA1-10) (23 nM 5(6)-FAM/OD_600_) (Tukey HSD, *p* < 0.05). To confirm that this mutant functions in other Pseudomonads, the experimental setup was also analyzed in *P. aeruginosa*. In this host, the (G3RQ) mutant considerably reduced the leaky expression from the phi15 RNAP to 29 5(6)-FAM/OD_600_ upon induction with 5 mM rhamnose, a remarkable 13-fold lower compared to the wild-type phi15 lysozyme (one-sided Student’s *t*-test, *p* < 0.01) (Figure 7c).

The results also indicate that the third position in the amino acid sequence plays an important role in RNAP inhibition and that a charged amino acid (R,Q) is preferred over the small glycine residue in the phi15 lysozyme sequence to create a strong interaction with the phi15 RNAP. These results correspond to previous work where point mutations in the N-terminal region of the T7 lysozyme caused a lack of RNAP inhibition, thus showing that even single point mutations can significantly impact the lysozyme–RNAP interaction [32]. 

Lastly, it can be noted that there are also large differences in the msfGFP output between the strains in the uninduced condition (Figure 7b), which could be attributed to the minor leaky expression of the lysozymes from the RhaRS/*p_rhaBAD_* system [18,34]. Overall, these results suggest that the other phage lysozymes could also be optimized for reduced toxicity and increased RNAP inhibition in a similar manner to enable tightly-controlled expression systems for SynBio applications.

### 2.6. Flow Cytometry-Based Quantitative Assessment of the phi15 Expression System

The phi15 RNAP and phi15 lysozyme (G3RQ) form a stringent expression system in *P. putida* and *P. aeruginosa* together with the phi15 promoter (Figure 7b,c). To characterize this system on a single-cell level, a flow cytometry experiment was performed on the *P. putida* and *P. aeruginosa* wild-type strains, the strains with the phi15 RNAP, phi15 lysozyme, and reporter construct, and the strains with the phi15 RNAP, phi15 lysozyme (G3RQ), and reporter construct (Table 1).

All of the *P. putida* and *P. aeruginosa* samples displayed single, homogenous populations, indicating that most of the cells responded to the presence of the inducers in a similar manner, with a limited occurrence of escapers. In addition, the wild-type controls showed very little response to the inducers in terms of the FITC-A (related to msfGFP expression). This allowed us to determine the threshold of the background FITC-A for *P. putida* and *P. aeruginosa* in this experiment, which was set at 10^4^. The results of the phi15 wild-type and phi15 (G3RQ) strains support the observations made in the previous spectrophotometric data (Figure 6). Both in *P. putida* and *P. aeruginosa*, the presence of phi15 lys (G3RQ) reduced the median FITC-A and the size of the induced cell population, resulting in a significantly improved fold induction in comparison to the wild-type phi15 lysozyme (fold induction 9.64 vs. 4.10 for *P. putida* and 129.78 vs. 7.36 for *P. aeruginosa*, respectively) (Table 1). Remarkably, in *P. aeruginosa*, the phi15 lysozyme G3RQ reduced the FITC-A, even below the value observed for the wild-type strain (FITC-A 527 vs. 647, respectively). As such, these results confirm the superiority of the phi15 lysozyme (G3RQ) over its wild-type counterpart and illustrate the potential of the phi15 expression system as a tool in *P. putida* and *P. aeruginosa*.

## 3. Discussion

Despite the widespread applications of the T7 transcriptional machinery in *E. coli* [35], its implementation in the *Pseudomonas* species has remained restricted due to the troublesome cytotoxicity of the T7 RNAP in this genus [7,14,15,16,17], as observed in this study as well. In this work, we mined four viral RNAPs from T7-related *Pseudomonas* phages, phi15, PPPL-1, Pf-10, and 67PfluR64PP, of which none impacted the fitness of SynBio host *P. putida*. In addition, these RNAPs displayed a broad range of transcriptional activity, a high level of orthogonality towards each other, and orthogonality to the host RNAP. This is in contrast to minor T7 promoter recognition that was observed for the host machinery. Two of the phage RNAPs, phi15 and PPPL-1 RNAP, also showed significant activity in *P. aeruginosa*, while no transcriptional activity was observed from the Pf-10 and 67PfluR64PP RNAPs in this host. The reason for this inactivity remains unclear but could potentially be due to improper expression of the *RNAP* genes, as they were not codon-optimized to the host.

Although the T7-like RNAPs did not display the same extremely high transcriptional activity as the T7 RNAP control, we argue that this should not be considered a disadvantage. Their non-toxicity, full host orthogonality, and more balanced activity allow easier cloning and flexibility toward different experimental setups compared to their T7 counterpart [14,15]. Furthermore, this mini orthogonal RNAP library enables considered selection of an RNAP for a specific application, in which phi15 and PPPL-1 RNAP are more suited towards applications that require high-yield of the gene of interest, while Pf-10 and 67PluR64PP RNAP enable more tightly-regulated and balanced expression of toxic intermediates or end products. For example, high amounts of the fluorinase enzyme are required for in vivo biofluorination with *P. putida* [36], whereas overexpression of the glycolate oxidase enzyme in *P. putida* allows efficient ethylene glycol conversion into polyhydroxyalkanoates [37]. On the other hand, tight expression control is preferred for endolysin expression for the controlled cell lysis of putida [38] and the study of toxic phage proteins in *P. aeruginosa* [39]. Furthermore, the orthogonal RNAP library allows the creation of various AND gates, OR gates, and resource allocators [6,8]. Due to the modularity of the T7-like RNAPs, these enzymes can be split into an enzymatic module and a promoter-recognition module. An AND gate is created by placing the two modules under the control of different inducible promoters and the desired output under the phage promoter, which only yields the output when both inducers are present. In addition, the enzymatic module of one RNAP can be paired with the promoter-recognition module of other phage RNAPs, thus enabling the creation of multiple AND gates in parallel, which all rely on the same core module. This concept was coined as a resource allocator, as the total amount of output solely depends on the core module and does not increase and overburden the cell when multiple promoter-recognition modules are expressed simultaneously. Thirdly, the T7 and phi15 RNAPs can be assembled into an AND gate in combination with the T7 promoter. When either the T7 or the phi15 RNAP are expressed, the desired output will be produced—though in lower amounts by the phi15 RNAP. 

The remarkable transcriptional activity of viral RNAPs often leads to high levels of leaky expression of the gene of interest under uninduced conditions. We successfully tackled this problem by introducing the corresponding phage lysozymes in the expression hosts, mirroring the proven strategy of the T7 system [3]. Interestingly, while the phage RNAPs showed high specificity towards their native phage promoter, the phage lysozymes proved to be much more promiscuous. Indeed, the Pf-10 lysozyme reduced leakiness from the phi15 RNAP by 84%, whereas the native phi15 lysozyme only showed a 30% reduction in leaky msfGFP expression. These surprising results inspired a directed mutation analysis of the phi15 lysozyme for improved RNAP inhibition, leading to the engineering of the high-performant, non-toxic phi15 lysozyme (G3RQ) mutant. This result is consistent with the work by Cheng et al. [32], where an R3S mutation in the T7 lysozyme reduced the T7 RNAP inhibitory effect. In addition, the performance of the 67PfluR64PP and PPPL-1 lysozymes could be improved to reduce the basal expression from their corresponding RNAPs. Furthermore, the toxicity observed upon overexpression of the phage lysozymes can be alleviated by knocking out the amidase activity of these enzymes, as this activity is the likely source of the observed toxicity [32,40].

Overall, this work provides a set of non-toxic, orthogonal viral RNAPs with well-defined promoter sequences and lysozyme-based RNAP repressors for the *Pseudomonas* species to expand the SynBio toolbox of this genus and allow the design of a plethora of synthetic genetic circuitry, inspired by years of T7-based research. In future work, the potential of these phage-based expression systems can be further optimized and validated with proofs-of-concept with industrially relevant compounds. Several improvements could be investigated, including increased genomic stability with genomic integration of the phage RNAP and an optimized and standardized reporter construct with reliable promoter variants and potent transcriptional terminators, as well as tackling potential protein aggregation issues [17,21].

## 4. Materials and Methods

### 4.1. Bacteriophage Genomes

All bacteriophage sequences used in this work originated from phages that were isolated, sequenced, and annotated in previous research, as shown in Table 2.

### 4.2. Bacterial Manipulation

In this study, two *E. coli* strains were used for vector cloning purposes, i.e., *E. coli* TOP10 as a main host and *E. coli* PIR2 for pBGDes-derived vectors carrying the R6K origin (Appendix A). The characterization and optimization of phage-based elements were performed in *P. putida* KT2440 or *P. aeruginosa* PAO1 (Appendix A). All strains were cultured overnight in a sterile LB medium or LB agar, supplemented with antibiotics as required: Amp^100^, Kan^50^, Gm^10^ (*E. coli* and *P. putida*) or Gm^30^ (*P. aeruginosa*), Tc^10^ (*E. coli* and *P. putida*) or Tc^60^ (*P. aeruginosa*), and Sp^50^ and Sm^200^. *E. coli* and *P. aeruginosa* were incubated at 37 °C, whereas *P. putida* was standardly incubated at 30 °C. Plasmid vectors were introduced in all strains by transformation. *E. coli* was transformed using the rubidium chloride method [44], whereas *P. putida* and *P. aeruginosa* were electroporated as described elsewhere [45]. The pBGDes vectors were always co-electroporated with a helper plasmid, pTNS2, to ensure genomic integration of pBGDes in the Tn7 *attB* site of the host. All strains used in this work are listed in Appendix A.

### 4.3. Vector Construction

To screen the selected phage RNAPs, promoters, and lysozymes to create a tailored pET system for *P. putida*, the SEVAtile vector set was used, which enables rapid and standardized assembly of genetic circuits [34]. As a positive control, the T7-based pET system was recreated with the SEVAtile vectors in *P. putida* and *P. aeruginosa*, as shown previously, with the T7 RNAP in pSTDesX, the T7 lysozyme in pSTDesR, and a reporter construct with *P_T7,MCP_-msfGFP* integrated into the Tn7 *attB* site using pBGDes [34]. To screen phage RNAPs, promoters, and lysozymes from T7-related phages in a similar setup, all necessary vectors were assembled using SEVAtile-shuffling by first amplifying the phage-encode parts with a tail-PCR to add the required overhangs for SEVAtile-shuffling [34]. Gibson assembly was used for vector assembly in case the phage genes contained one or multiple BsaI recognition sites, as indicated in Appendix A. Assembled vectors were introduced in *E. coli* TOP10 or *E. coli* PIR2 for pBGDes-derived vectors, and the correct insertion of phage-encoded genes was verified by Sanger sequencing (Eurofins Genomics, Ebersberg, Germany). All used primers and vectors of this work are listed in Appendix A. 

### 4.4. Toxicity Evaluation by Growth Curve Monitoring

To assess the potential cytotoxicity of recombinantly expressed phage RNAPs and lysozymes on *P. putida* and *P. aeruginosa*, 12 h growth curves of all relevant cultures were prepared. First, overnight cultures of four biological replicates were diluted 20-fold in a fresh growth medium in a 96-well plate with the appropriate antibiotics and incubated for 3 h while shaking at the appropriate temperature. At this time point, every cell culture was split in an uninduced and induced fraction by adding the appropriate inducer to the latter. For RNAP toxicity, a final concentration of 1 mM 3-methylbenzoate (3mBz) was introduced, while for lysozyme toxicity, 10 mM L-rhamnose was supplied to the culture. Culture plates were directly placed in a CLARIOstar^®^
*Plus* Microplate Reader (BMG Labtech, Ortenberg, Germany), where OD_600_ measurements were performed every 30 min for a total period of 12 h while incubating at the appropriate temperature with intermittent shaking. The resulting data were corrected for blank values (sterile growth medium) and statistically analyzed using JMP 16 Pro (JMP^®^, Version 16. SAS Institute Inc., Cary, NC, USA, 1989–2021). Multiple comparisons of the mean values were performed on the final timepoint by first confirming the normality of the data for each sample (Shapiro–Wilk test, α = 0.05), followed by the Tukey HSD (honest significant difference) test, with correction for multiple comparisons (α = 0.05).

### 4.5. Fluorescence Intensity Assays

To verify the performance of the phage elements in both *P. putida* KT2440 and *P. aeruginosa* PAO1, fluorescent expression assays were performed [34]. Overnight cultures of four biological replicates of *P. putida* KT2440 or *P. aeruginosa* PAO1 carrying the appropriate vectors were prepared in an M9 minimal medium containing 1× M9 salts (BD Biosciences, Franklin Lakes, NJ, USA), 0.2% citrate (Sigma Aldrich, St. Louis, MO, USA), 2 mM MgSO_4_ (Sigma Aldrich), 0.1 mM CaCl_2_ (Sigma Aldrich), 0.5% casein amino acids (LabM; Neogen^®^ Company, Lansing, MI, USA), and the appropriate antibiotics. Each overnight culture was diluted 20-fold in a fresh M9 medium in a Corning^®^ 96-Well Black Polystyrene Microplate with a Clear Flat Bottom and incubated for 2 h in shaking conditions. At this point, the cell cultures were split in two to create an uninduced and induced sample, to which the required inducer(s) was added. Next, the fluorescence intensity and OD_600_ levels were monitored every 30 min for 12 h on a CLARIOstar^®^ Plus Microplate Reader while incubating at 30 °C or 37 °C for *P. putida* KT2440 or *P. aeruginosa* PAO1, respectively. The fluorescent intensity of the msfGFP was measured at a 485 nm excitation wavelength and 528 nm emission wavelength with the enhanced dynamic range setting of the apparatus. All relative fluorescent measurements were blank-corrected for a sterile medium and normalized for cell growth by dividing by the corresponding OD_600_ value. To convert the relative fluorescence units of the msfGFP to absolute units, a calibration curve was added to each experiment [46]. More specifically, 0, 375, 750, and 1500 nM of 5(6)-carboxyfluorescein (5(6)-FAM) (Sigma Aldrich) in phosphate-buffered saline was added to each plate in duplicate. All (normalized) fluorescent measurements of the msfGFP were subsequently converted to the equivalent 5(6)-FAM concentration. The data were analyzed, visualized, and verified for statistical significance using JMP 16 Pro. Statistical significance assays were performed on the final timepoint by first confirming the normality of the data for each sample (Shapiro–Wilk), followed by an appropriate mean (multiple) comparisons test. In particular, if the data were normally distributed, a (pairwise) Student’s *t*-test (α = 0.05) was performed, while for non-normally distributed data, a (pairwise) Wilcoxon assay (α = 0.05) was employed. No corrections for multiple comparisons were made due to large differences in variance between samples, except for the lysozyme optimization assays for inhibition of the phi15 RNAP, where variances were equal (Tukey HSD test, α = 0.05).

### 4.6. Transcription Start Site Determination with 5′-Capping-RACE

The transcription start site (TSS) of each phage promoter was determined using 5′-capping-RACE (Rapid Amplification of cDNA Ends) [27]. First, the total RNA fraction of *P. putida* strains, pA0RA0, pB0RB0, pC0RC0, pD0RD0, and pE0RE0, was harvested as follows (Appendix A). Overnight cultures were diluted 100-fold in a fresh LB medium with appropriate antibiotics and incubated at 30 °C in shaking conditions. Once the cells reached OD_600_ 0.3, cultures were induced with 0.3 mM 3mBz and incubated for 3 h upon harvesting at OD_600_ 4. The harvested cells were subjected to the hot phenol/lysozyme method to extract total RNA [47], followed by DNase I treatment. Next, cDNA was generated according to the protocol described by Liu et al. (2018) with the primers listed in Table 3. The resulting cDNA products were cloned into pSTEntry with SapI restriction–ligation and transformed to *E. coli* TOP10. Five transformants of each sample were treated with a GeneJET Plasmid Miniprep kit (Thermo Scientific) to isolate the pSTEntry.phage-cDNA vectors and Sanger sequenced with SEVA_PS1 and SEVA_PS2 primers (Appendix A).

### 4.7. Flow Cytometry

Single-cell fluorescent data of strains were obtained by flow cytometry. First, overnight cultures were prepared in duplo, which was diluted 20-fold in a fresh M9 medium the following day in a clear 96-well plate with a flat bottom and incubated for 2 h in shaking conditions. At this point, the cell cultures were split in two to create an uninduced and induced sample, to which the required inducer(s) was added. After overnight induction, samples were diluted tenfold in 200 µL of a PBS medium (pH 7.4, filter-sterilized (0.22 µm)) and analyzed on a CytoFLEXS^®^ Flow Cytometry machine (Beckman, San Jose, CA, USA). Then, 5000 events (i.e., individual cells) were screened for FSC-A (gain 165), SSC-A (gain 400), and FITC-A (gain 10), with a maximal flow rate of 1000 events/µL. The FITC-A channel detected msfGFP fluorescence of single cells, where a value above 10^4^ was considered positive for msfGFP fluorescence.

## Figures and Tables

**Figure 1 ijms-24-07175-f001:**
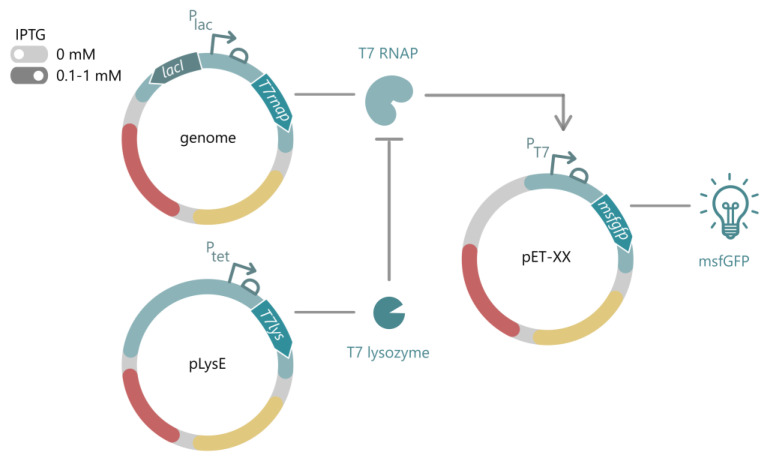
General layout of the T7-based pET system for *E. coli*. The T7 RNAP is stably integrated into the host genome with an IPTG-inducible expression cassette. In the absence of IPTG (light grey), the system is considered off, while upon addition of IPTG (dark grey, usually 0.1–1 mM IPTG), the T7 RNAP is expressed from the LacI/*Plac* system. The T7 RNAP drives expression of a gene of interest, here depicted as an *msfGFP*, from its putative T7 promoter on any pET vector. To express toxic proteins, alternative hosts are available, carrying the pLys vector for expression of the T7 lysozyme, which inhibits transcriptional activity of the T7 RNAP in uninduced conditions. IPTG: isopropyl-β-D-thiogalactopyranoside, T7 RNAP: T7 RNA polymerase, msfGFP: monomeric superfolder green fluorescent protein.

**Figure 2 ijms-24-07175-f002:**
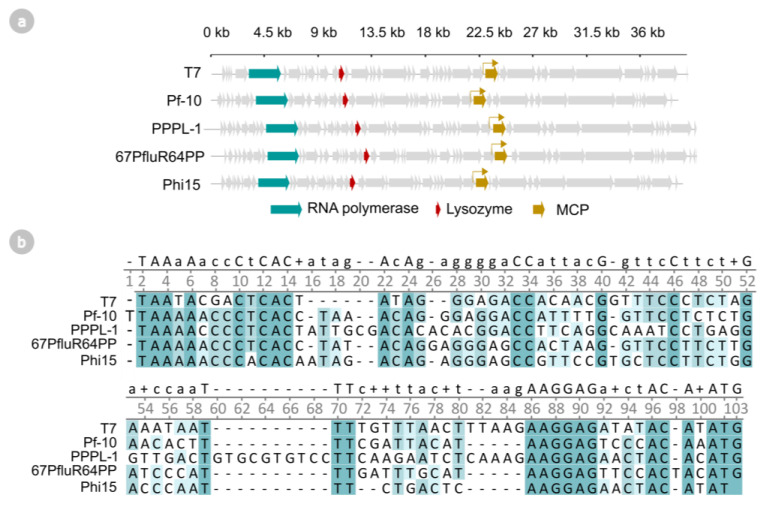
(**a**) Genomic organization of phages T7, Pf-10, PPPL-1, 67PfluR64PP, and phi15. The RNA polymerase genes, lysozyme genes, and major capsid protein (MCP) genes are indicated in teal, red, and yellow, respectively. (**b**) Multiple sequence alignment (ClustalOmega) of the predicted phage promoter and 5′ untranslated region (UTR) of the phage’s major capsid protein. Darker shaded background colors indicate a higher level of conservation.

**Figure 3 ijms-24-07175-f003:**
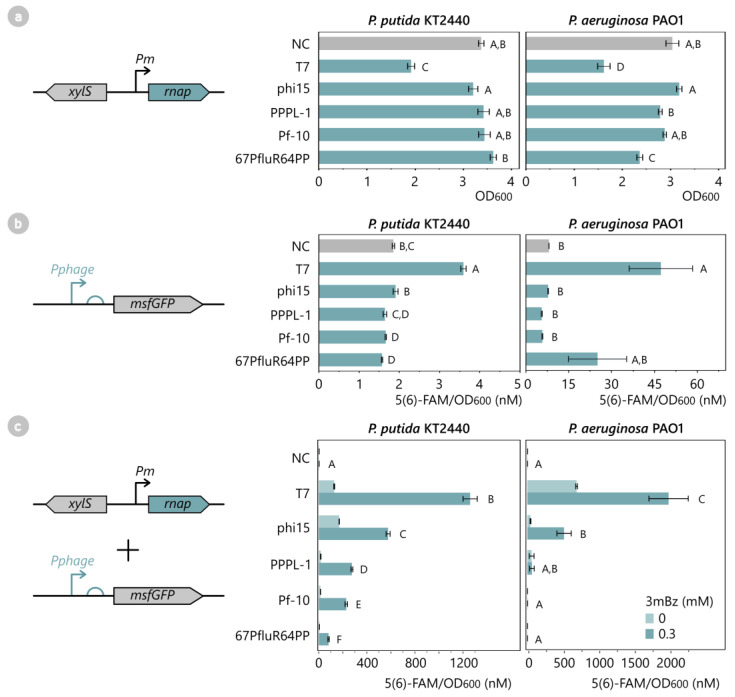
(**a**) Effect of phage RNAP expression on cell growth of *P. putida* KT2440 and *P. aeruginosa* PAO1. The RNAPs of phages T7, phi15, PPPL-1, Pf-10, and 67PfluR64PP were introduced in *P. putida* and *P. aeruginosa*, as well as an empty pSTDesX vector as a negative control (NC). All samples were induced with 1 mM 3mBz at OD_600_ 0.3 for 12 h. Bars and error bars represent the mean OD_600_ and standard error of four biological replicates after 12 h induction, respectively. Samples not connected by the same letters are significantly different (Tukey HSD, α = 0.05). (**b**) Recognition of the phage promoter by the host RNAP of *P. putida* KT2440 and *P. aeruginosa* PAO1. MsfGFP reporter constructs with the phage-specific promoters of T7, phi15, PPPL-1, Pf-10, and 67PfluR64PP and a promoterless construct (NC) were introduced in *P. putida* or *P. aeruginosa* and were monitored for 12 h for OD_600_ and msfGFP levels. As the phage RNAPs were not present, no msfGFP expression was expected unless the phage promoter was also recognized by the host RNAP. Bars and error bars represent the mean value and standard error of four biological replicates after 12 h cell growth, respectively. Samples not connected by the same letters are significantly different (Tukey HSD, α = 0.05). (**c**) T7-like phage RNAPs generate high msfGFP expression levels from their putative phage promoter in *P. putida* KT2440 and *P. aeruginosa* PAO1. All phage RNAPs and corresponding *phage promoter-msfGFP* reporter constructs, including empty control vectors (NC), were introduced in *P. putida* and *P. aeruginosa* and were induced with 0.3 mM 3mBz for a 12 h period. The fluorescent intensity was normalized for OD and expressed as an equivalent 5(6)-FAM concentration (nM). Bars and error bars represent the mean value and standard error of four biological replicates after 12 h induction, respectively. Samples not connected by the same letters are significantly different (Wilcoxon (*P. putida*) and Student’s *t*-test (*P. aeruginosa*), α = 0.05). The complete growth curves and fluorescence expression levels over time of (**a**–**c**) are provided in Appendix A.

**Figure 4 ijms-24-07175-f004:**
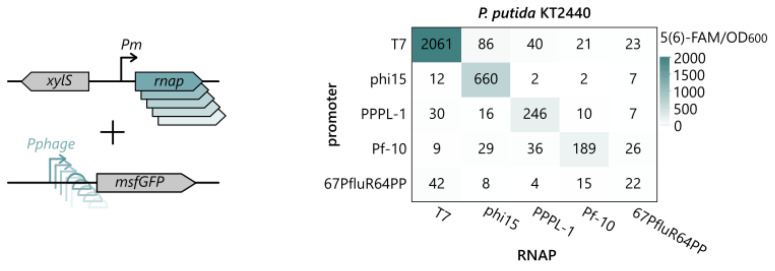
To verify cross-recognition between the phage RNAPs and their promoters, all 25 combinations of RNAPs and *phage promoter-msfGFP* reporter constructs were introduced in *P. putida* KT2440 and induced with 0.3 mM 3mBz overnight. The fluorescent intensity was normalized for the OD and expressed as an equivalent 5(6)-FAM concentration (nM). Values represent the mean normalized fluorescence intensity after overnight induction of four biological replicates. The statistical analysis of all pairwise comparisons and a connecting letter report are provided in Appendix A.

**Figure 5 ijms-24-07175-f005:**
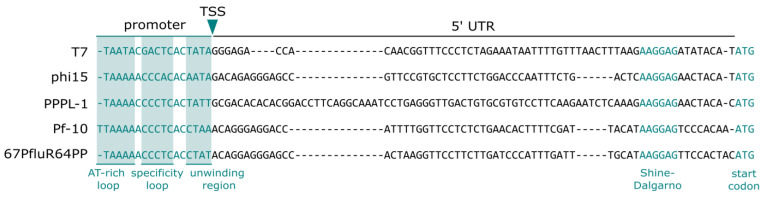
Clustal-omega alignment of the validated promoter and 5′ UTR of the phages’ major capsid protein, separated by the confirmed transcription start site (TSS). Confirmed promoter regions of the T7 promoter and 5′ UTR, namely the AT-rich recognition loop, specificity loop, unwinding region, Shine–Dalgarno sequence, and start codon, are projected on the promoters of the other phages.

**Figure 6 ijms-24-07175-f006:**
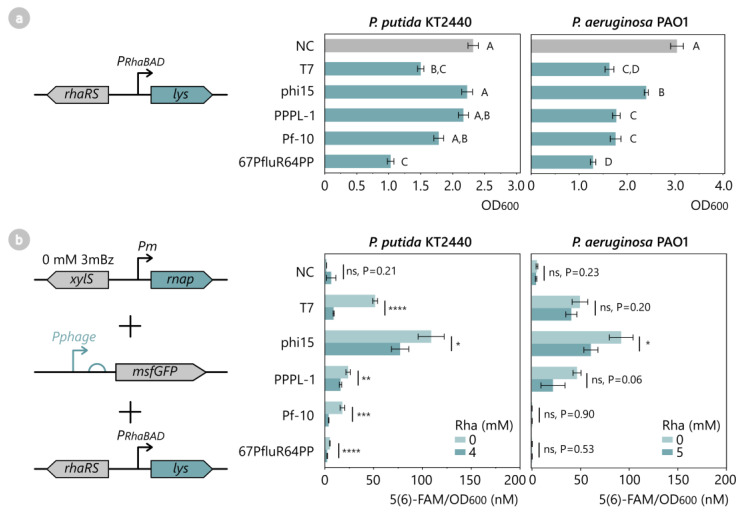
(**a**) Toxicity assay of phage lysozymes in *P. putida* KT2440 and *P. aeruginosa* PAO1. All phage lysozymes of T7, phi15, PPPL-1, Pf-10, and 67PfluR64PP, as well as an empty pSTDesR control vector (NC), were introduced in *P. putida* and *P. aeruginosa* and were induced with 10 mM Rha at OD_600_ 0.3, after which cell growth was monitored for 12 h. Bars and error bars represent the mean and standard error of four biological replicates. Samples not connected by the same letters are significantly different (Tukey HSD, α = 0.05). (**b**) Fluorescence assay to analyze the inhibitory effect of the phage lysozyme on its corresponding phage RNAP. All phage RNAPs, lysozymes, and *phage promoter-msfGFP* reporter constructs of T7, phi15, PPPL-1, Pf-10, and 67PfluR64PP and empty control vectors (NC) were introduced in *P. putida* and induced with 4–5 mM Rha for 12 h. Data points represent the mean 5(6)-FAM/OD_600_ value of four biological replicates, while error bars represent the standard error. Significance levels are indicated for one-sided Student’s *t*-tests, with *: *p* < 0.05, **: *p* < 0.01, ***: *p* < 0.001 and ****: *p* < 0.0001. Full growth curves and fluorescence expression levels over time of (**a**,**b**) are available in Appendix A.

**Figure 7 ijms-24-07175-f007:**
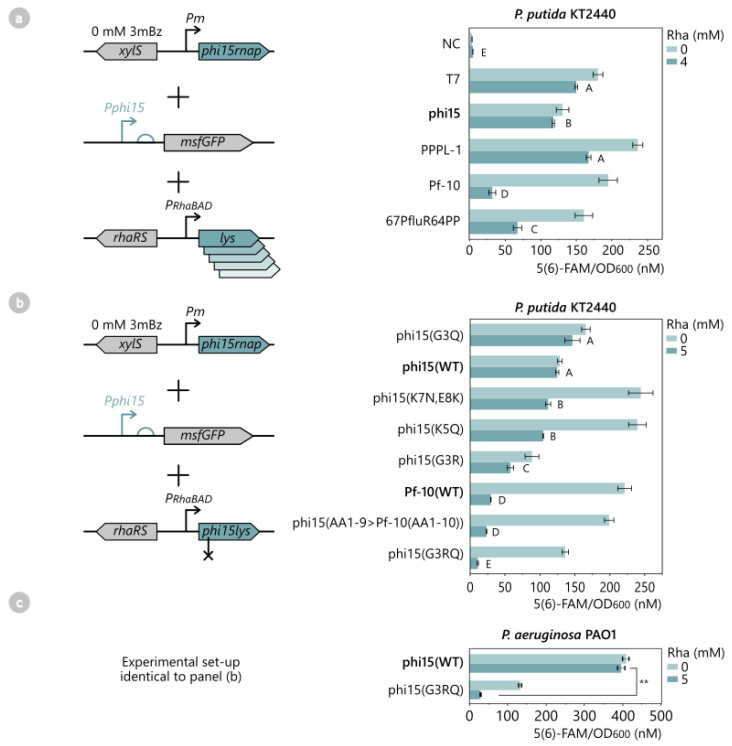
(**a**) Fluorescence intensity assay to analyze the inhibitory effect of different phage lysozymes on the phi15 RNAP. The phage lysozymes of T7, phi15, PPPL-1, Pf-10, 67PfluR64PP, and an empty pSTDes3 control vector (NC) were introduced in *P. putida* KT2440 together with the phi15 RNAP and Pphi15-msfGFP reporter construct. All samples were induced with 4 mM Rha at OD_600_ 0.3, after which the fluorescence intensity and cell growth were monitored every half hour for 12 h. Bars and error bars represent the mean 5(6)-FAM/OD_600_ value and standard error of four biological replicates, respectively. Samples not connected by the same letters are significantly different (Tukey HSD, α = 0.05). (**b**) Similar fluorescence assay as (**a**) to assess the inhibitory strength of different phi15 lysozyme mutants on the phi15 RNAP. The wild-type phi15 and Pf-10 lysozymes (WT), as well as phi15 lysozyme mutants (AA1-9 > Pf10(AA1-10)), (G3R), (G3Q), (G3RQ), (K5Q), and (K7N,E8K), were introduced in *P. putida*, together with the phi15 RNAP and Pphi15-msfGFP reporter construct. All samples were induced with 5 mM Rha for 12 h. Bars and error bars represent the mean 5(6)-FAM/OD_600_ value and standard error of four biological replicates, respectively. Samples not connected by the same letters are significantly different (Tukey HSD, α = 0.05). (**c**) Assessment of the inhibitory strength of the phi15 lysozyme (G3RQ) mutant in *P. aeruginosa* PAO1. The wild-type phi15 lysozyme (WT) and phi15 lysozyme mutant (G3RQ) were introduced in *P. aeruginosa*, together with the phi15 RNAP and Pphi15-msfGFP reporter construct, and induced with 5 mM Rha for 12 h. Bars and error bars represent the mean 5(6)-FAM/OD_600_ value and standard error of four biological replicates, respectively. The significance level for a one-sided Student’s *t*-test is indicated, with **: *p* < 0.01. Full growth curves and fluorescence expression levels over time of (**a**–**c**) are available in Appendix A.

**Table 1 ijms-24-07175-t001:** Flow cytometry data of the phi15 expression system in *P. putida* and *P. aeruginosa*. Wild-type *P. putida* KT2440 and *P. aeruginosa* PAO1 strains (wild-types), *P. putida* and *P. aeruginosa* with the phi15 RNAP, phi15 reporter construct, and phi15 lysozyme (phi15), and *P. putida* and *P. aeruginosa* with the phi15 RNAP, phi15 reporter construct, and phi15 lysozyme (G3RQ) mutant (phi15(G3RQ)) were induced overnight with 5 mM Rha (+lys) or 0.3 mM 3mBz (+RNAP), after which 5000 cells were analyzed with flow cytometry for FITC-A, as described in Section 4. Cells with a FITC-A level above 10^4^ are considered induced, whereas cells below 10^4^ are uninduced. Column FITC-A depicts the median FITC-A value of the entire cell population, and column induced (%) depicts the percentage of cells of the entire population that have a FITC-A value above 10^4^. Complete histograms of the corresponding data are available in Appendix A.

		***P. putida* Wild-Type**		***P. putida* phi15**		***P. putida* phi15 (G3RQ)**	
		FITC-A	Induced (%)	FITC-A	Induced (%)	FITC-A	Induced (%)
−RNAP	+lys	44	0.16	17,803	80.00	12,730	66.66
−RNAP	−lys	70	0.12	15,270	74.74	32,128	78.50
+RNAP	−lys	99	0.30	72,956	95.40	122,739	92.82
Fold induction *		2.25		4.10		9.64	
		***P. aeruginosa* Wild-Type**		***P. aeruginosa* phi15**		***P. aeruginosa* phi15 (G3RQ)**	
		FITC-A	Induced (%)	FITC-A	Induced (%)	FITC-A	Induced (%)
−RNAP	+lys	647	3.22	33,429	77.52	527	5.94
−RNAP	−lys	692	6.18	7820	47.74	18,672	81.10
+RNAP	−lys	559	3.22	245,893	90.98	68,392	74.58
Fold induction *		0.86		7.36		129.78	

* Fold induction is the ratio of FITC-A of the (−RNAP, +lys) sample and the (+RNAP, −lys) sample.

**Table 2 ijms-24-07175-t002:** List of bacteriophage genomes used in this work.

Bacteriophage	Accession Number	Reference
Phi15	FR823298.1	[41]
Pf-10	NC_027292.1	Unpublished
PPPL-1	NC_028661.1	[42]
67PfluR64PP	MH179478.2	[43]

**Table 3 ijms-24-07175-t003:** Primers used to determine the transcription start site of phage promoters with 5′-capping-RACE.

Name	Sequence
TSS_TSO	ACACTCTTTCCCTACACGACGCTCTTCCGATCTrGrGrG
TSS_outerprimer	AATGATACGGCGACCACCGAGATCTACACTCTTTCCCTACACGACGCTCTTCCGATCT
TSS_innerprimer	ATAGCTCTTCTAGACTACACGACGCTCTTCCGATCT
TSS_GSP1	TCAGTTTACCGTTGGTTGCATCACCTTCACCTTCACCACGAACAGAGAATTTGTGGCC
TSS_GSP2	TAGGCTCTTCTCTTCGAACAGAGAATTTGTGGCC

## Data Availability

The data presented in this study are available in the main article and Appendix A.

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
