# Peer review of "Assessing the Orthogonality of Phage-Encoded RNA Polymerases for Tailored Synthetic Biology Applications in Pseudomonas Species"

_ijms, 2023, doi:10.3390/ijms24087175_

Round 1
Reviewer 1 Report
To find an appropriate “pET expression system” for Pseudomonas, the authors investigate toxicity and expression of T7-like phage RNA polymerases and their associated lysozymes. The investigated phages are chosen as they do infect Pseudomonas species, utilising the fact that these phages already successfully can replicate within Pseudomonas. By cloning RNAPs into the bacteria of interest, as well as the lysozyme, toxicity and efficiency is evaluated, leading to the conclusion that these RNAPs work as an expression system, with the benefit of providing diversity in terms of amount of expression and stability. Overall, I find this paper relevant and nicely written, the work behind it is thorough and extensive but still summarised in a nice way. There are some points that I think need to be adressed, see the attachment.

Author Response
We are grateful to the reviewers for their thorough review and careful consideration of our manuscript. Based on their remarks and suggestions, we believe we could considerably improve the manuscript in both content and style. All requested changes of both reviewers have been implemented. Moreover, inspired by one comment, we have introduced a new flow cytometry experiment to provide single cell quantitative data on selected strain populations, which provides in depth information on the population behavior upon expression of the phi15 RNAP and lysozyme. Below, we provide a detailed discussion on all changes that were made to the manuscript, in response to the reviewers’ remarks.
Reviewer 1
The introduction is nicely written and gives a good understanding of the system and where it originates from. However, while the authors bring up the fact that other bacteria would be valuable for biotechnology, they do not give any reasoning for this. Would other bacteria expand the uses of SynBio? Would other bacteria perform better? Is it economical interest behind it? I would appreciate some more words regarding why this paper is needed, which I do not doubt it is, just to make it clearer.
Thank you for your words of support! Indeed, in some cases other bacteria are more adept for the production of compounds than E. coli, due to differences in metabolism, resistance to toxic intermediates or end-products or their ability to thrive in the specific production conditions. We included this statement in the introduction for clarification.
The structure of the result section would benefit from a reorganisation, either to fit the figures that are presented or to rearrange the figures to fit the text.
The figures were restructured to retain all figures and their corresponding legend on the same page and to fit better within the text.
The result section is quite long, and sometimes a bit hard to follow with regards to abbreviations of clone variants, while the discussion is very brief. This gives focus and frames the points of the paper in a nice way, but there are points that are not discussed (line 363- 364) or only briefly mentioned (lines 369-372) without proper examples. Also, the AND and OR gates are mentioned in the introduction and then hinted on in the results, but not explained anywhere or discussed.
The discussion section was expanded significantly, with a brief discussion for original lines 363-364, concrete examples for original lines 369-372 and an explanation of possible AND and OR gates that can be constructed with the identified RNAPs.
Figure 1 is a nice figure, but the figure text could be more expansive. It is unclear what the colours for IPTG indicate and some of the abbreviations would be clearer if expanded, e.g. mentioning msfGFP and what function it has in the pET system.
Thank you for indicating that the abbreviations are not trivial to all readers. We expanded the figure legend and included explanations for the used abbreviations.
Line 59-60, the pET system originates from genes from the T7 phage which infects E. coli, I find the phrasing “applied towards” somehow misses the evolutionary background of this phage-host pair and instead suggests that the system was bioengineered and then tweaked for E. coli.
We concur and adjusted the phrasing to “the pET system is based on coliphage T7 and was further optimized towards E. coli”.
Line 111. “and msfGFP”, this is not explained and an explanation would facilitate understanding of the results.
We included the brief statement “as the phage RNAPs are not present, no msfGFP expression is expected, unless the phage promoter is also recognized by the host RNAP” for clarification.
Line 297-298, if nine amino acids were exchanged for ten amino acids, how many amino acids of the original protein remained?
The original protein consists of 154 amino acids, of which the first nine N-terminal AAs were substituted for the first ten AAs of Pf-10 lysozyme. The other 145 amino acids remained unchanged.
Line 306-309, these abbreviations (and similar ones above) are hard to tease apart, and when then relying on them to pinpoint functionality it is hard to follow. I would appreciate either an explanation to the rationale behind the abbreviations or a list in supplemental where each abbreviation is described in more detail, I see table S3 might contain this but it is again hard to follow.
The abbreviations were removed from the text and replaced with brief explanations of the strains to improve the flow of the text.
Several “Error! Reference source not found” occur in the material and methods.
We apologize for the inconvenience and fixed the error messages in the manuscript.
Lines 444-446 and 470-472 referring to “appropriate tests” should be expanded, what tests, how did you decide what tests where appropriate, in what software was the tests done, etc.
The tests used for each assay were previously mentioned in the main text and figure legends of the original version of the manuscript. We now also included additional information in the method section on which tests where used for normally and non-normally distributed data, as well as the significance levels used.
Supplementary still contains some track changes.
It appears we did not upload the final version of the supplementary information and sincerely apologize for this mistake.
Figure S2 comes before Figure S1 in the text.
The numbering has now been updated.
Figure S2 – what is meant by “tree scale 1”? The length of that seem to be greater than the radius of the tree?
Thank you for bringing this mistake to our attention. The tree scale now has the correct size and represents the phylogenetic distance between samples. This explanation has also been added to the figure legend.
Figure 3 comes before Figure 2 in the text, Figure 2b is also referred to before Figure 2 is referred to (and Figure 2a is never mentioned in the text).
We added an additional referral to Figure 2 in section 2.1.
Figure S3 – the different parts of this figure are named a-c and 3, but in the figure text referred to as “1-4”. “b” has no label on the y-axis.
We corrected the labelling in both the text and figure.

Reviewer 2 Report
The authors in the paper investigate functional RNA polymerases applicable to Pseudomonas species. The experimental methods satisfactorily support the conclusions of the paper. The research is basic but provides an important foundation to establish synthetic genetic circuits in Pseudomonas. I have a few important comments that I would like the authors to address.
-I would encourage the authors to cite other references and discuss any other important RNAP elements that have been reported to be functional in Pseudomonas for the benefit of the readers so as to provide a background of existing knowledge.
-Please include fluorescence images of the bacteria to qualitatively compare the fluorescence intensity.
Author Response
The authors in the paper investigate functional RNA polymerases applicable to Pseudomonas species. The experimental methods satisfactorily support the conclusions of the paper. The research is basic but provides an important foundation to establish synthetic genetic circuits in Pseudomonas. I have a few important comments that I would like the authors to address.
We thank the reviewer for their kind words!
I would encourage the authors to cite other references and discuss any other important RNAP elements that have been reported to be functional in Pseudomonas for the benefit of the readers so as to provide a background of existing knowledge.
We included lines L77-81 in the introduction to provide a concise overview of the main research performed on this topic and as such hope to provide the reader with sufficient background to this work.
Please include fluorescence images of the bacteria to qualitatively compare the fluorescence intensity.
It is not clear to us whether the reviewer is referring to microscopic or other fluorescent images of the bacteria. Regardless, we believe that qualitive fluorescent images would not hold much additional value to the presented data.
However, the request is valid and therefore we have performed a new experiment, introducing flow cytometric data of selected strains. This cytometric data provides additional, quantitative information on the single cell behavior and population dynamics of the samples.
The M&M (lines 580 to 590) and results & discussion for this section can be found on lines 386 to 419 and in newly introduced table 1 and Supplementary figures S10 and S11.

Round 2
Reviewer 2 Report
I thank the authors for addressing my questions/comments.
Author Response
We thank the reviewer for their time and consideration of our manuscript.